# Development and Validation of Standardized Quality of Life Measures for Persons with IDD

**DOI:** 10.3390/bs13060452

**Published:** 2023-05-30

**Authors:** Antonio M. Amor, Miguel Á. Verdugo, María Fernández, Alba Aza, Victoria Sánchez-Gómez, Zofia Wolowiec

**Affiliations:** 1Institute for Community Inclusion, University of Salamanca, 37005 Salamanca, Spain; verdugo@usal.es (M.Á.V.); mariafernandez@usal.es (M.F.); azhernandez@usal.es (A.A.); vsanchezgomez@usal.es (V.S.-G.); 2Department of Personality, Assessment, and Psychological Treatments, Faculty of Psychology, University of Salamanca, 37005 Salamanca, Spain; 3Department of Radiology, University Hospital of Salamanca, 37007 Salamanca, Spain; zwolowiec@saludcastillayleon.es

**Keywords:** quality of life, assessment, standardized instruments, validation, rights, supports

## Abstract

The implications of the individual quality of life (QoL) model of Schalock and Verdugo have made it the most cited QoL model in the field of disability. The QoL model is understood as a conceptual and applied framework for action that allows the materialization of the rights of persons with disabilities through the multidimensional assessment of these persons using QoL indicators, and the development of actions guided by these values and supported by evidence. The purpose of this work is to present the foundations of this model and offer a step-by-step guide to developing standardized QoL assessment instruments and providing evidence that supports their use to implement the model in practice. This paper explores relevant topics such as: (a) the need to identify critical population groups and contexts; (b) the identification of QoL indicators for said groups and contexts; (c) the development of items focused on the assessment of personal outcomes; (d) provision to the items of validity evidence based on content and pilot measure design and (e) validation process to gather evidence that supports the uses of the instrument. Last, a framework that allows using the evidence on personal outcomes as disaggregated and aggregated data at different levels of the social system is presented, thus highlighting the role of the model as a change agent regarding individuals, organizations and schools, and public policy.

## 1. Introduction

Over the last two decades, different legal documents have been passed to safeguard the rights of persons with disabilities. The main milestone in this matter occurred in 2006, when the United Nations Convention on the Rights of Persons with Disabilities (UNCRPD) and its optional protocol were enacted, both entering into force on 3 May 2008. With it, the State parties committed to initiating legal reforms to comply with the social and civil rights enshrined in the document.

Although hand in hand with the signing and ratification of treaties such as the UNCRPD [1] there has been an increase in the social inclusion of persons with disabilities, these changes have not occurred in a linear fashion and situations such as the COVID-19 pandemic have produced a setback in the rights of persons with disabilities [2,3]. Moreover, as different authors appoint (e.g., refs. [4,5]), the signing of documents and treaties such as the UNCRPD is important, but it does not translate into a real improvement in the access and enjoyment of rights for persons with disabilities, nor does it mean that the way supports are delivered is beneficial to persons with disabilities (e.g., sometimes supports can be intrusive and/or be based on the availability of resources rather than on the self-determination of individuals).

Therefore, the need to commit to models that allow the values contained in the UNCRPD [1] to be translated into practice is clearly shown. In this sense, the quality of life (QoL) model of Schalock and Verdugo [6] is commonly seen as a reference framework for addressing this need, given its conceptual and practical implications. This model is the most accepted QoL model in the disability field [7,8] since it does not only provide a conceptual basis to understand what QoL is, but it also offers a framework for action that allows the materialization of the rights of persons with disabilities through the multidimensional assessment of said persons using QoL indicators, and the development of actions guided by these values and supported by evidence gathered from assessment [9]. The purpose of this work is twofold: (a) Explaining the conceptual and applied underpinnings of the QoL model and (b) offering a step-by-step guide to developing standardized QoL assessment instruments and providing evidence that supports their use to implement the model in practice.

## 2. The Quality of Life Model

### 2.1. The Concept of Quality of Life

From the perspective of Schalock and Verdugo’s model [6], individual QoL is defined as a state of personal wellbeing that: (a) Is multidimensional, that is, it is composed of eight essential domains in the lives of all people (i.e., emotional wellbeing [EW], physical wellbeing [PW], material wellbeing [MW], personal development [PD], interpersonal relationships [IR], social inclusion [SI], rights [RI], and self-determination [SD]); (b) has universal and cultural properties; (c) has objective and subjective components and (d) is influenced by personal and environmental factors, as well as by the interaction between them.

The QoL model also shares the same set of values about persons with disabilities that underlies the UNCRPD [1]: Equity, equality, empowerment, and support. This model already proposed, four years before the UNCRPD was passed, rights as a key domain in the lives of persons with disabilities [6]. In a recent work by Amor et al. [9], the authors of the model highlighted its key conceptual foundations: (a) Respect for the persons and the domains that make up their lives; (b) respect for their rights and self-determination and (c) to seek the satisfaction of their aspirations and needs based in personalized and person-centered approaches in all relevant contexts.

### 2.2. The Quality of Life Model as an Applied Framework: Measurement Framework and Change Agent

Beyond the conceptual bases of the model, it is necessary to understand how to articulate the proposal in a systematic framework for action that allows transferring the values shared with the UNCRPD to the measurement and improvement of the QoL and rights of persons with disabilities [9]. Moreover, this is possible because the model stands as a measurement framework and as a change agent, as subsequently described in the next sections.

#### 2.2.1. Quality of Life Model as a Measurement Framework

The QoL domains are operationalized through core indicators [10,11] which are defined as perceptions, behaviors, or specific conditions of the QoL domains that reflect personal wellbeing, and that are observable and measurable [6]. Although the QoL domains are universal (i.e., cross-cultural) [12,13], the indicators are sensitive to the culture and the characteristics of a given group. That is, while QoL domains are the same for all people, QoL core indicators vary from culture to culture, from group to group, and from context to context [14]. As an example, for both a Polish child without disabilities and a Spanish child with intellectual and developmental disabilities (IDD), EW is an essential domain regarding their QoL, but the elements that are indicative of EW are not the same for each one, since Poland and Spain are not the same cultures (e.g., different policies and attitudes towards persons with disabilities), and the support needs of typically developing children and children with IDD differ. This characteristic of the indicators entails that they must be identified and follow a validation process for specific cultures, groups, and contexts.

After identifying the core indicators, these are developed through items that allow measuring personal outcomes (defined as aspirations and needs of the person in the core indicators of the domains) thus monitoring the individual’s QoL [11]. The operationalization of the QoL domains through their core indicators, and items, is essential to implement the model in practice as a measurement framework and, therefore, as a change agent.

A final characteristic of the model as a measurement framework is its alignment through the core QoL indicators, with the rights enshrined in the UNCRPD [15,16]. In this sense, the validation process of the indicators, and the assessment of personal outcomes beyond measuring QoL, is a way of knowing the state of the individuals in relation to the access and enjoyment of their rights [14]. Table 1 provides a summary of the QoL domains and their description, the core indicators identified for people with IDD, and the UNCRPD rights aligned with the core indicators of each domain.

The operationalization of the QoL domains through their indicators and the importance of measuring personal outcomes have resulted in the development of multiple standardized tools designed to monitor the QoL of people and design and implement personalized supports focused on achieving personal-desired goals that allow persons with disabilities to enjoy their rights and improve their QoL [9]. Table 2 lists the different standardized QoL assessment tools that have been developed in Spain based on this model, highlighting the target population and the QoL assessment approach taken.

Some of these tools have undergone adaptation and validation processes in different countries, such as the INICO-FEAPS in Colombia [29] and the San Martín and KidsLife scales in the United States [30].

#### 2.2.2. Quality of Life Model as a Change Agent

The information obtained through the assessment of personal outcomes using the tools described above can be analyzed as disaggregate (i.e., at the individual level) or aggregate data, thus allowing the collection of different types of information to be used according to the goal of the QoL evaluation. This analysis of QoL scores can be used to support evidence-based decision-making at different levels [31]. The use of personal outcome evidence to support decision-making processes make QoL a change agent [6].

Understanding the role of the QoL model as a change agent makes it necessary to underline another characteristic of the model: QoL is based on a systems perspective [32]. This perspective assumes that people live in a complex social system made up of different levels (i.e., microsystem, mesosystem, and macrosystem) that encompass the necessary areas for people to live, develop, and have the opportunity to improve, and that influence the development of people’s values, beliefs, behaviors, and attitudes, thus affecting their QoL [6]. The microsystem refers to the immediate context in which the person lives and that directly affects the person (e.g., living placement, family, or friends). The mesosystem includes everything that directly affects the functioning of the microsystem (e.g., neighborhood, community, organizations, or schools). Finally, the macrosystem alludes to the broader cultural patterns, sociopolitical trends, and economic factors that directly affect values and beliefs. Some examples of the use of personal outcomes assessed through standardized measures at these levels can be found in social services [33,34]. These uses include the planning of personalized supports, comparisons between organizations, and the definition and assessment of public policy. Recent trends have started to arise claiming the use of the implications of the model regarding educational systems [35]. Information about the uses of the evidence on personal outcomes at different levels of the social system is provided in the last section. Figure 1 represents Schalock and Verdugo’s QoL model as a conceptual and applied framework from a systems perspective.

## 3. Development and Validation of Quality of Life Standardized Measures

As stated in the previous section, indicators are sensitive to both the characteristics of specific groups and contexts where these groups live. Therefore, translating this model into practice through the development of standardized QoL assessment measures requires following a series of cumulative steps.

The first step will always be the identification of a vulnerable population group along with the context in where this target group participates. Radiography of both elements is critical for identifying core QoL indicators for said group and context. Without this, it is not possible to design a standardized QoL measurement instrument. It is worth stressing again that while the QoL domains are universal, indicators are not, and indicators vary not only from group to group but also from context to context. For example, imagine that a research team is focused on children with IDD. The research team knows for sure that the relevant QoL indicators for this group are not the same as for adults with physical disabilities. However, what they must have clear is that for this group different contexts will make different indicators relevant. Thus, if the focus is to develop a standardized measurement instrument aimed at children with IDD who go to special schools, the core indicators will be different than those relevant for children with IDD enrolled in general schools. The reason for this is that the contextual elements involving both types of settings are different (e.g., different schools’ cultures, policies, and practices). This all affects the QoL of the target group and the development of standardized QoL measures must take this into account. Often, identifying target groups and contexts is not difficult, in the sense that vulnerable groups and the contexts these groups take part in are well known by relevant stakeholders. The rest of the steps necessary to address the development and validation of standardized QoL assessment tools are depicted in Figure 2.

Prior to explaining the steps involved in the validation of QoL assessment tools, it is necessary to stress the two assessment approaches followed under Schalock and Verdugo’s QoL model [6] and comment on the implications that this has for both the structure of the tools and for the validation steps. QoL assessment can follow an approach based on the information provided by third parties (i.e., report of others) or by the person whose QoL is of interest (i.e., self-report) [10]. Logically, when planning the development of a QoL assessment standardized instrument, depending on the population, the research team may select one approach or another (or both of them). Typically, for children under the age of 12, reporting of others is preferred, while for adults, self-report is more common. However, this is not always the case because it also depends, among other variables, on cognitive functioning (something that must be taken into account when developing standardized QoL measures for individuals with IDD given their significant limitations in both intellectual functioning and adaptive behavior). The criteria to select one or another are beyond the scope of this paper. Those interested in this topic, please go to the work by Balboni et al. [36].

In the report of others, the tool is implemented through a semi-structured interview by a qualified interviewer who knows Schalock and Verdugo’s QoL model and who is familiar with the tool itself and with the principles of educational/clinical interview. The interview will be conducted with an informant or observer (or a dyad of informants), who is a relative or a professional who knows the person whose QoL we want to assess for at least three months and who has had recent opportunities to observe the person in different domains. The person to be assessed is a person who pertains to the target group (e.g., a student with IDD attending general education). The task of the interviewer is to use the items of the instrument as an interview script to ask the informants about the QoL of the person of interest and fill the instrument. Conversely, self-report assessment implies providing the instrument to the person whose QoL is going to be assessed, and the person reads and answers the items (self-administration), or answers the questions asked by an interviewer who fills the tool (interview). Notwithstanding the assessment approach, the structure of QoL assessment instruments is the same regarding their sections (i.e., information and sociodemographic data, and assessment scale), although there are slight differences in the content of the sections depending on the assessment approach.

In the information and sociodemographic data section, the relevant information is collected regarding the interviewer, the informants, and the person being evaluated (i.e., report of others), or just regarding the person being assessed and the interviewer if necessary (i.e., report of others). In addition to sociodemographic data, it is important to collect information regarding other relevant variables (e.g., clinical or educational) that help to understand the QoL scores obtained. This information is essential to understand which personal and environmental factors are associated with better or worse QoL scores to make the best decisions. The main section of any standardized QoL instrument is the QoL assessment scale. This section is typically subdivided into eight subsections or subscales, each one corresponding with a QoL domain and including between 8 and 12 items aimed at assessing personal outcomes for each domain. No matter the assessment approach, each item is rated on a four-point frequency rating scale from 1 = Never to 4 = Always. As stated above, in the report of others, the items are used by the interviewer to obtain information from the observers about the person’s QoL, while in the self-report, the information is directly provided by the person. For the case of reports of others and self-reports that involve an interview, the interviewer must select the response option that best describes the thoughts and feelings shared by the informants, while in self-administered instruments, the person whose QoL is of interest is responsible to fill the instrument (with the required support when necessary). The language also varies from the report of others (third-person singular) to self-report versions (first-person singular).

Retaking the steps involved in the validation (Figure 2), the assessment approach has implications regarding the ethical principles concerning participants. These differences are commented on below. 

### 3.1. Step 1. Guaranteeing the Ethical Principles of Research

Every study requires compliance with ethical principles, especially if it implies the access, collection, storage, treatment, and analysis of personal data. There are different ethical declarations of application in different fields that establish a series of basic guidelines (e.g., the Declaration of Helsinki of 1964 and its amendments). Regardless of them, the ethical principles in the process of psychometric validation of standardized QoL assessment tools always require, at a minimum (see, e.g., [14]): (a) Having a clear data processing protocol that ensures the anonymity of the participants; (b) protecting the data and preventing loss of sensitive information (e.g., having a good server to store the information); (c) informing all the participants (by oral and written explanation) about the purpose of the instrument and the research, and about their rights and obligations as participants and (d) distributing and collecting signed informed consent forms from all the participants. This step is essential and neglecting it may result in legal liability. For this reason, apart from the requirements presented here, it is mandatory that those interested in validating QoL assessment tools contact ethics committees to ascertain the requisites that this kind of research entails.

The development of standardized QoL assessment tools implies access to participants at three different moments (steps two, three, and four in Figure 2). In this sense, it is necessary to start working on the ethics principles application as soon as the need to develop a standardized QoL assessment instrument is identified. As mentioned, the assessment approach has implications regarding ethical principles. In step four, when conducting the field test validation for the cases when a report of others is going to be developed, it is necessary to count on consent from both external informants (observers) and from the persons whose QoL is of interest. On the other hand, when self-report assessment tools are developed, only informed consent forms from the persons being evaluated are needed. Special emphasis must be given when working with underage persons and persons whose legal capacity has been modified (e.g., adults with IDD and extensive and pervasive support needs). Once the research team has received approval from the research ethics board, then the validation process can be addressed. 

Guaranteeing ethical principles goes beyond formal documents. In the last two decades, a debate has arisen about the need of including persons with disabilities as active agents over the research that affects them and not only consider them as subjects of study, especially regarding individuals with IDD. Although it is not the goal of this work, it is necessary to recognize the importance of maintaining a balance between rights, consent, benefits to be generated in the lives of people with disabilities and their participation as active agents in research that concerns them. Those readers interested in this subject will find relevant information in this regard in the following sources [37,38,39,40,41]. 

### 3.2. Step 2. Identifying the Core Quality of Life Indicators and Relevant Items for the Group and Context of Interest

Determining the core indicators for the group and context of interest usually implies two stages: (a) two parallel reviews, of which one is a literature review on QoL + the population of interest + the target context and a review of already-existing assessment tools for the target population and context and (b) a discussion between relevant stakeholders.

The reviews are aimed at identifying an initial set of QoL indicators (and items) for the group and context of interest. The literature review must be conducted using the main databases related to the field of study (e.g., QoL and students with IDD using ERIC, PsycInfo, PsycArticles, Academic Search Complete, Psychology and Behavioral Sciences Collection, or Medline) and relevant search terms (a thesaurus can help). Although English is the main *lingua franca* in research, complementing English-language searches using local languages and databases is an optimal way to reach out to additional relevant literature regarding the group and context of interest (e.g., complementing the search on QoL and students with IDD using Spanish search terms in databases such as Redalyc or Scielo-Spanish edition). Notwithstanding the language and the databases consulted, the search should be focused on both environmental and personal characteristics, to identify relevant indicators for each QoL domain (i.e., indicators associated with the condition of interest and with the contexts where the people with such condition participate). Beyond identifying QoL indicators, the literature review is also useful to start defining a preliminary set of items describing QoL aspirations and needs relating to the indicators. The initial set of items can be fed if, for the group and context of interest, there are other assessment instruments aimed at QoL or similar constructs (e.g., health-related QoL tools for PW). As a result of this stage, a starting set of QoL indicators and an initial pool of items (20–30 per domain, depending on the number of indicators) can be achieved (e.g., ref. [14]).

At a later stage, it is necessary to hold a discussion between relevant stakeholders (i.e., direct-practice professionals, researchers, family members, and persons with the condition of interest) to see the relevance of this preliminary set of items, taking as a reference the purpose of the instrument that wants to be validated and the QoL indicators identified. As an output of this second step, a pool of items describing personal outcomes for the core indicators that compose the QoL domains for the target group and context is obtained, and this pool of items needs to follow a validation process regarding its content.

### 3.3. Providing the Items with Validity Evidence Based on Content

Years of experience of the research team in the development and validation of standardized QoL assessment tools have made the team follow the Delphi method to assess the extent to which the items identified in the previous step show evidence of content validity, that is, if they reflect the universe of aspirations and needs relevant for the indicators and domains in the context and group of interest (e.g., refs. [14,42,43]). The Delphi method is an information-gathering technique that allows consensus on the opinion of participants through repeated consultations. These participants are experts in the group and context of the study, and their expertise can be both personal (i.e., relative or person with the condition of interest) and/or professional (e.g., researcher or direct-care professional). Delphi studies must take care of the number of participants. Thus, there has to be a sufficient number of participants to reflect all stakeholders and, at the same time, not be too large for everyone to participate [44]. Two decades of experience of the research team in the validation of standardized QoL assessment instruments have made the research team conduct four-round Delphi studies to address the analysis of the evidence on the content validity of the initial pool of items. Although it is important to acknowledge that Delphi studies do not follow a closed scheme or a predetermined number of rounds (rather, they end when an agreement is reached [44]), the research team takes this approach because using such a number of rounds allows participants to take different roles regarding the initial pool of items by completing different tasks (e.g., assessment of already-generated items, generation of new items, assessment of items created by colleagues, debates with colleagues, etc.). A brief comment on each round is provided below. For detailed examples of Delphi studies in the field of validation of standardized QoL instruments, please go to [14,42,43].

In round one after receiving training in QoL and the tool that is being developed, participants have to undertake two tasks. The first task consists in assessing the content of the items. For this purpose, all the participants have to evaluate each item against four criteria: (a) Suitability (the extent to which, in their opinion, the item belongs to the QoL domain where it was presented to them); (b) importance (relevance of the item to assess the QoL domain in the target group and context); (c) observability (if the item describes a situation that is observable by an external agent) and (d) sensitivity (whether the item reflects a situation that can be changed because of implementing supports). To assess each criterion, participants have a four-point Likert-type rating scale (1 = Not suitable/important/observable/sensitive to 4 = Completely suitable/important/observable/sensitive). The items with an *M* ≥ 3.5 and an *SD* < 1 for all of the four criteria are retained, as this is indicative that they adequately represent personal outcomes for the target group and context. In the second task, participants are asked to propose a maximum of five new items per domain. The research team decides which of the new items are incorporated into the pool after applying qualitative criteria [45].

In the second round, participants have to decide whether to incorporate the items that they previously generated in the second task of Round one and that were considered appropriate by the research team. In this case, only the items that gather the agreement of 12 out of 15 participants are included in the pool.

Round three typically focuses on the QoL domains that have retained the fewest number of items. In this case, participants discuss the relevance of retrieving the discarded items for these QoL domains, and they can propose new items. Finally, the research team summarizes all the information and presents it to participants to reach an agreement on the final number of items that are added to these domains.

Round four is similar to the first task of Round one. Participants must assess the suitability, importance, observability, and sensitivity of the items incorporated in Rounds two and three using the same procedure, rating scale, and decision-making as in task one of Round one. With this, a final pool of suitable, important, observable, and sensitive items (i.e., with evidence on content validity) is available. These items constitute the pilot QoL instrument that will continue the validation process through the field test validation.

Last, to ensure the quality of the process followed and the agreement of the participants when judging the items using each criterion, Bangdiwala’s weighted statistic (B^W^_N_) is calculated [46] for this final pool of items. This is a statistic that expresses, from 0 to 1, the strength of the participants’ agreement when they are judging the suitability, importance, observability, and sensitivity of the items (the closer it is to 1, the greater intensity of the agreement).

Another way to generate items can be through focus and discussion groups. The crucial question, however, beyond the strategy to identify the items, is to provide them with valid evidence based on content, and this section has provided a how-to framework to help in this task.

### 3.4. Field Test Validation

The output of step three is the development of a field test version of the standardized QoL assessment tool that will continue the validation process. The structure of the field test version will vary slightly depending on the assessment approach. The field test versions of the standardized measures include another section in addition to the explained ones (i.e., information and sociodemographic data, and QoL assessment scale). This additional section gathers identification data of participants and ensures anonymity using alphanumeric codes that identify all parts involved in the research (i.e., the interviewer, the informant/observer if necessary, and the person being evaluated). A field test validation follows different phases: (a) Selection of participants; (b) pilot study and (c) implementation of the instrument and analyses of its psychometric properties.

#### 3.4.1. Selection of Participants

It is necessary to clarify the participants that will constitute the sample for the test administration. In this sense, the sample should be representative of the population group (in the context of interest) [47]. The representativeness of the sample entails taking into account both the sample size and the proportionality regarding the distribution of relevant variables such as gender or clinical characteristics [48].

When calculating the sample size, it is necessary to think about people whose QoL is going to be assessed through the instrument and not in terms of the informants/observers (especially, if the tool follows a report of others). There are different formulas to calculate a minimum sample size. Of the different proposals, the most used is that developed by Cochran [48]. To calculate a sample size representative of a large population, this author proposed the following formula:(1)$$n_{0}=\frac{Z^{2}\times p\times q}{e^{2}}$$
where: *n*_0_ = is the sample size to determine; *z* = critical confidence level; *p* = estimated proportion of an attribute that is present in the population; *q* = 1 − *p* = desired level of accuracy (as a default value, following the maximum uncertainty principle *p* = *q* = 0.5). Imagine we would like to determine a sample size of an infinite population (*N* ≥ 100,000) whose extent of variability is unknown. Assuming the maximum uncertainty principle and a 95% confidence level with a 5% of accuracy, the calculation of the sample size would be the following:(2)$$n_{0}=\frac{1.96^{2}\times (0.5)\times (0.5)}{{\left(0.05\right)}^{2}}=384.16=385\;participants$$

Given that in the development of standardized QoL assessment instruments researchers work with specific groups in contexts that are very well defined (e.g., adults with IDD and extensive and pervasive support needs), it is common to work with populations *N* < 100,000 whose number is known by researchers. Thus, it is necessary to implement a variation of Equation (1) to determine an optimum sample size for such cases. Cochran [48] addressed such adaptation:(3)$$n=\frac{n_{0}}{1+\frac{\left(n_{0}-1\right)}{N}}$$
where: *n*_0_ = sample size determined following Equation (1) and *N* is the (known) population size. To illustrate this formula with an example, imagine that a research team wants to validate a standardized QoL assessment instrument in Spain for students with IDD aged 6–12 years enrolled in general education settings. The research team knows that for these age bands and contexts, the population of students with IDD is *N* = 42,824. For determining the minimum sample size, the research team has previously calculated *n*_0_ following Equation (1) and obtained the value of Equation (2) *n*_0_ = 385. Now that the research team has *n*_0_, it is time to substitute the *N* in Equation (3):(4)$$n=\frac{385}{1+\frac{\left(385-1\right)}{42,824}}=381.58=382\;participants$$

In this case, given that the population size is large, there is not a big difference between formulae (the difference becomes more evident the smaller the population size). Default values are usually taken to solve Equation (1) thus affecting Equation (3). It is not the purpose of this work to enter long-standing discussions regarding the assumptions underlying these default values and the implications they have. Interested readers will find a relevant debate on this matter in [49,50]. Other authors (e.g., [51]), to avoid the use of the previous formulae, propose other alternatives, such as including, for each item of the pilot instrument, between five and 10 participants. Another alternative focuses on guaranteeing, at least, 200 participants. However, this is not exempt from controversy and there is no agreement between the different authors [47].

As mentioned, to be representative of the population the sample needs to be stratified according to relevant variables. Regarding the different sampling strategies, the stratified random sampling strategy is considered the best one in terms of representativeness. Although this is the ideal, it is not always possible to achieve, especially in the field of disability (very specific population groups and contexts are difficult to access). Therefore, the use of incidental sampling after calculating the minimum sample size and taking into consideration strata is a common way to address representativeness, although limitations in terms of generalizability are expected [48].

#### 3.4.2. Pilot Study

Next, it is necessary to conduct a pilot study with a set of participants who have similar characteristics to those that constitute the sample for the test administration. Pilot studies aim to assess whether, in a similar but smaller sample, participants understand the items and whether the instrument is working properly. These preliminary studies allow detecting, avoiding, and correcting possible errors at an early stage, as well as seeing the functioning of the tool from an applied point of view [47]. Any issue during the tool administration (e.g., excessive complexity of the administration process, items that are not well understood, etc.) should be reported. Next, it is necessary to conduct preliminary analyses regarding the psychometric quality of the items (e.g., discrimination index, factor loadings or differential functioning) and the evidence of validity and reliability.

#### 3.4.3. Administration of the Instrument and Analyses of Its Psychometric Properties

Once the pilot study has been conducted and the changes (if any) derived from it have been incorporated, it is time to access all the participants required, implement the instrument, and collect data for the evaluation of its psychometric properties. Although a psychometric validation process must take into consideration the different sources of validity and reliability, this article lists the analyses that are considered essential for the psychometric validation of these types of measures, based on two decades of rigorous research [14,42,43,52]:The psychometric properties of the items are used to select a maximum number of items with the best properties for each QoL domain. For this, five criteria are considered [42]: (a) The mean value of the scores for each of the items and their standard deviation; (b) the number of missing data; (c) the corrected homogeneity indexes; (d) the distribution of the responses and (e) the content of the items.Regarding the analyses of the evidence of validity and reliability, validity analyses must focus, above all, on validity evidence based on internal structure (i.e., analyzing the structure of the tool using confirmatory factor analysis) and on validity evidence in relation to other variables. Reliability must consider at least an internal consistency coefficient (e.g., Cronbach’s alpha, ordinal alpha, or Omega coefficient) and, if possible, the inter-rater reliability (e.g., through intraclass correlation coefficient) [53,54,55,56,57,58].

Table 3, created after analyzing relevant publications focused on standards in psychometric validation processes [53,54,55,56,57,58], describes the main sources of validity and reliability and provides foundations for interpreting the analyses and the evidence found:

### 3.5. Publication of the Tool

Once the items and the evidence of validity and reliability have been studied, then the best decisions can be made to choose the best items and the structure of the final version of the tool [52]. Finally, there will be a dully calibrated standardized instrument (with its own norms and user manual) that can be used to measure personal outcomes. Strategies to assess personal outcomes and uses of the information are succinctly explained below.

## 4. Using the Tool: Assessment of Personal Outcomes and Uses of the Information Gathered

Once the calibrated instrument is available, it is necessary to understand its uses to bring the QoL model into practice. In this sense, these standardized measures can be applied directly to persons with disabilities at the individual level (microsystem). In this case, QoL assessment serves different purposes. For example, it is possible to detect the state of the aspirations and needs of individuals in each domain. With it, it is possible within frameworks such as person-centered planning, to design and implement personalized supports focused on the achievement of desired life goals which, in turn, translate into the enhancement of the QoL of the individuals and greater enjoyment of their rights [9]. At the same time, QoL assessment can be used to inform which support strategies lead to better personal outcomes after interventions. Another key element is the fact that these tools are standardized. This allows, for example, comparing the QoL scores of two persons to shed light on comparative QoL-related needs and aspirations [27].

Beyond the uses at the microsystem level, aggregate data (i.e., high-level data obtained by combining individuals’ QoL scores) can be used both at meso- and macrosystem levels. At the mesosystem level, the use of aggregate data of organizations’ users allows the ongoing improvement of the organizational quality and the redefinition of the organizations, and it also allows for establishing the profiles of suppliers and decision-making in relation to the improvement of the plans implemented to enhance their outputs. The comparative analysis of the aggregate data provides information on the performance of a given organization (i.e., analyzes how these aggregate data vary over time) and facilitates the comparison between different organizations. With the information shared between organizations, it is possible to detect organizational strategies that have been of help to enhance the QoL of users and propose these strategies as examples of best practices to address common challenges. Finally, regarding the macrosystem, the implementation of the QoL concept allows for guiding new ways of developing, implementing, monitoring, and evaluating public policies and social initiatives [31].

## 5. Conclusions

In this work, the QoL model of Schalock and Verdugo [6] has been presented as a framework to move from positive values to actions guided by these values and supported by evidence to enhance the QoL and enjoyment of the rights of persons with disabilities. Moreover, the study has offered a step-by-step guide to developing standardized QoL assessment tools and providing evidence that supports their use to implement the model. Beyond the information presented in this article, the development and validation of QoL assessment tools can be addressed in a different way. In this sense, it is possible to accomplish the translation and validation of already-existing instruments to contexts different from the original, as has already been done in the field of QoL and IDD [29,30]. Those interested in addressing this alternative may find the guidelines published by the International Test Commission interesting [59].

The implementation of the QoL model [6] using standardized measures in the field of disability is getting more and more support. For example, in the European context, the European Association of Service Providers for Persons with Disabilities (EASPD), which represents over 20,000 disability organizations across Europe, has opted for the use of one of the QoL assessment tools based on this model and aimed at students with IDD [14] to monitor the European Child Guarantee. The European Child Guarantee is a strategy focused on guaranteeing access to basic services for children in vulnerable situations. With it, the EASPD seeks to provide policymakers with a framework and a tool that help to gather evidence on the extent to which the aspirations and needs of the children in need are actually being covered by their national plans. The nature of Schalock and Verdugo’s QoL model will also allow for the development of local plans for countries that are still developing their national strategies (i.e., the indicators can help to identify aspirations and needs that should be targeted by the plans). Beyond the development of measurement instruments and their use, the future of the field lies in sharing experiences of the achievements obtained after applying the tools at the different levels of the system, which is the goal of the model: To enhance the lives of persons with disabilities [31].

## Figures and Tables

**Figure 1 behavsci-13-00452-f001:**
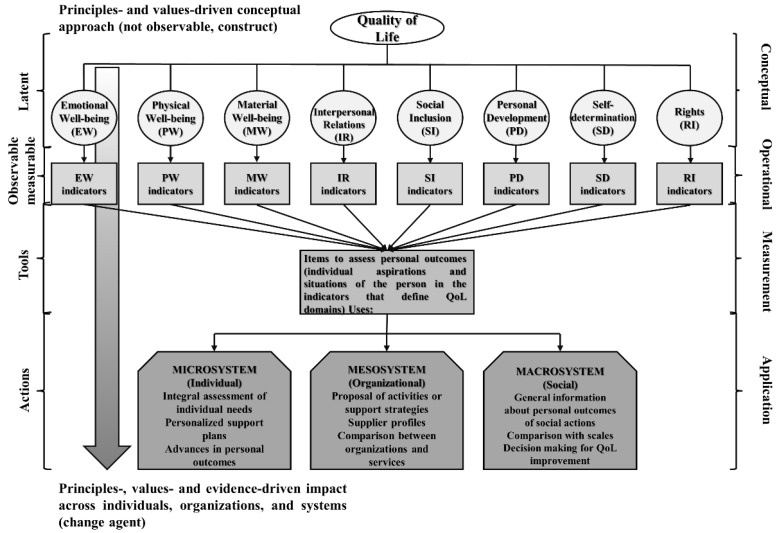
QoL Model as a conceptual and applied framework [9] (p. 71). The figure offers a visual impression of the transition from values to evidence-oriented actions focused on the improvement of individual’s QoL and enjoyment of rights. Ellipses and circles reflect conceptual aspects, while rectangles define specific actions (operationalization and measurement, and decision-making based on evidence).

**Figure 2 behavsci-13-00452-f002:**
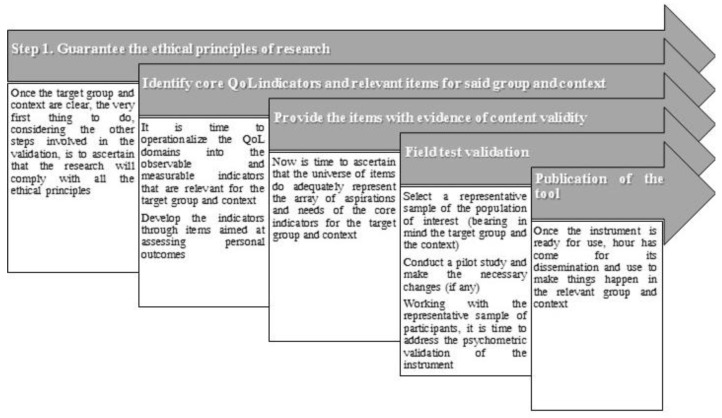
Steps involved in the validation of standardized QoL assessment instruments.

**Table 1 behavsci-13-00452-t001:** UNCRPD’s rights aligned with QoL’s measurement framework (self-elaboration based on [6,8,9,11,12,13,14]).

QoL Domain	Description	Indicators for People with IDD	Rights Aligned
EW	To be calm and safe, relaxed, and not to be overwhelmed and nervous	Satisfaction, self-concept, self-esteem, positive feelings, and lack of distress or negative feelings	Arts. 16 (freedom from exploitation, violence, and abuse) and 17 (protecting the integrity of the person)
IR	Having relations with different people, having clearly identified friends, and getting on well with others (acquaintances, neighbors, partners, etc.)	Social interactions, having identified friends, familiar interactions and relations, positive social contacts, relationships, communication, and sexuality	Art. 23 (respect for home and the family)
MW	Having enough money to buy whatever one needs and/or wants, having a proper household or workplace, having access to adequate services	Housing, workplace, employment status, salary (pension, income), belongings, savings, material goods, and access to services	Arts. 27 (work and employment) and 28 (adequate standard of living and social protection)
PD	Having the possibility of learning different things, accessing knowledge, developing new skills and personal competence (cognitive, social, and practical), and having the possibility of self-realization	Limitations/capacities, access to information and communication technologies, teaching-learning opportunities, educational status, work-related skills (or other activities), and functional abilities	Art. 24 (education)
PW	Being healthy, feeling fit, having good healthy habits	Health status, health care, healthy habits (e.g., rest and sleep, hygiene, eating or physical exercise), and activities of daily living (e.g., self-care, mobility)	Arts. 25 (health) and 26 (habilitation and rehabilitation)
SD	Being able to self-decide and having the opportunities to choose the things that one considers relevant according to one’s own values and beliefs, choosing one’s life, employment, leisure time, living, and the people to be with	Goals and personal values, decisions and choices, and autonomy/personal control	Arts. 14 (liberty and security of person) and 21 (freedom of expression and opinion, and access to information)
SI	Going to different places in the city or neighborhood where other people go, participating in different activities on equal foot with others, performing an active role in the community, and feeling part of society and having the support of others	Inclusion, participation, accessibility, supports, recognition, and community roles	Arts. 8 (awareness-raising), 9 (accessibility), 18 (liberty of movement and nationality), 19 (living independently and being included in the community), 20 (Personal mobility), 29 (participation in political and public life), and 30 (participation in cultural life, recreation, leisure and sport)
RI	Being considered and treated equally with other people, including having access to the same opportunities, being respected (i.e., personality, opinions, wishes, privacy, etc.), and knowledge of one’s own rights and exercise of them	Respect, intimacy, confidentiality, knowledge and exercise of rights (e.g., respect, dignity, equality), and legal guarantees (e.g., access or due process)	Arts. 5 (equality and non-discrimination), 6 (women with disabilities), 7 (children with disabilities), 10 (right to life), 11 (situations of risk and humanitarian emergencies), 12 (equal recognition before law), 13 (access to justice), 15 (freedom from torture or cruel, inhuman or degrading treatment or punishment), and 22 (respect for privacy)

EW = Emotional wellbeing; IR = Interpersonal relationships; MW = Material wellbeing; PD = Personal development; PW = Physical wellbeing; SD = Self-determination; SI = Social inclusion; RI = Rights.

**Table 2 behavsci-13-00452-t002:** Standardized QoL assessment instruments based on Schalock and Verdugo’s model developed in Spain.

Assessment Instrument	Target Group	Assessment Approach
CVI-CVIP: Quality of life Assessment Questionnaire in Childhood [17]	Children with and without special educational needs aged 8–11 years	Self-report and report of others
CCVA: Questionnaire for Assessing Quality of Life in Adolescent Students [18]	Adolescents with and without special educational needs between 12 and 18 years old	Report of others
KidsLife [19]	Children, adolescents, and youth with IDD	Report of others
KidsLife-Down [20]	Children, adolescents, and youth with Down syndrome	Report of others
KidsLife TEA [21]	Children, adolescents, and youth with autism spectrum disorder and ID	Report of others
FUMAT Scale [22]	Elderly persons recipient of social services	Report of others
GENCAT Scale [23]	Adults who receive social services	Report of others
Integral Scale [24]	Adults with IDD	Self-report and report of others
INICO-FEAPS Scale [25]	Adults with IDD	Self-report and report of others
San Martin Scale [26]	Adults with IDD and extensive and pervasive support needs	Report of others
CAVIDACE Scale [27]	Adults with brain injury	Report of others
CAVIDACE Scale—Self report version [28]	Adults with brain injury	Self-report
Quality of Life Index for Inclusive Education—Primary Education Version [14]	Students with IDD, behavioral and emotional concerns, and learning difficulties enrolled in primary, general education (6–12 years old)	Report of others

IDD = Intellectual and Developmental Disabilities; ID = Intellectual Disability.

**Table 3 behavsci-13-00452-t003:** Information regarding validity and reliability sources in validation studies (self-elaboration based on [53,54,55,56,57,58]).

Property	Source	Interpretation Foundations
Reliability: Consistency or stability of measurements when the measurement process is repeated	Internal consistency reliability: Extent to which the items correlate with one another	Strong correlations indicate that an assessment scale’s items have a robust relationship with one another and are, therefore, measuring different aspects of the same construct
	Split half reliability: Linear relationship between half of the items on a scale with the other half	A high correlation between the two halves suggests that items on the scale are measuring the same construct
	Test-retest reliability: Evaluates the consistency of a scale score over short periods of time	Strong correlations between scores from two separate and independent administrations completed following the same conditions at different time points suggest that the construct being assessed is stable
	Interrater reliability: Consistency of scale scores across assessors	If two separate and independent administrations of an assessment involve different evaluators and the correlations found between the scores in the two administrates are high, then the outcomes of the measure are trustworthy regardless of the administrator
Validity: Refers to the degree to which evidence and theory support interpretations of test scores for intended uses of the tests	Evidence based on content: The extent to which the items on an assessment adequately represent the universe of items that could be associated with the construct of interest	Content validity should be established when the measure is developed, when subscales are conceptualized, and items are written. Foundations are provided in Section 3.3
	Evidence based on internal structure: The degree to which the relationships between the items and test components conform to the construct on which the proposed interpretations of test scores are based	Support for a standardized instrument’s internal structure comes from research findings that demonstrate a strong relationship between the construct being measured and an assessment scale’s test items and the subscale scores. Statistical analyses revealing that the items share variance in ways that match the defined construct reflect positively on an instrument’s internal structure
	Evidence based on relation to other variables: The extent to which test-derived scores are related to measures of other variables that are theoretically associated (directly or inversely) with the construct assessed by the test	Evidence based on relation to other variables is established by collecting data that show that constructs that theoretically should be related are, in fact, related. The expected relationships may be of different types. For example, the traditionally called “convergent validity” focuses on convergent relationships. Convergent evidence provides support that a measure is correlated with other variables that claim to measure the same (or a similar) construct. Conversely, evidence on discriminate validity is established when gathering evidence that shows that constructs that theoretically should have no relationship with one another, in fact, have low correlations. In some cases, the instrument may be designed to predict a future characteristic or behavior. The test-criterion relationship may be assessed at the same time (concurrent), for example, by comparing two key groups that the instrument should identify as different, or by assessing the relationship of the instrument to a variable assessed at a later time (predictive)
	*External validity:* Refers to the extent to which results from a study using the tool can be generalized to other settings and people	It is important to assess the extent to which the scores obtained through the implementation of said instrument maintain evidence of good validity and reliability when applied to other settings and people, or in other countries and languages

## Data Availability

Not applicable.

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
