# Peer review of "Development and Validation of Standardized Quality of Life Measures for Persons with IDD"

_behavsci, 2023, doi:10.3390/bs13060452_

Round 1
Reviewer 1 Report
Table 1: I am finding important missing information in the description of the domains. Please refer to the shared indicators published in literature to base your domains descriptions (E.G.: in the description of SI it is missing the aspect of performing the same role, having an active role, as the other community members; RI: it is missing having access to the same opportunities as well. etc...).
line 318 Literature usually refers to a not fixed number of rounds for the Delphi, rounds terminate when agreements is found.
paragraph 3.4.3 There is a need to provide more references to support what is stated. Specially lines 452, 453, 454, 456.
line 89 and 90 need a formal English polishing
458 Typo
Author Response
Dear Reviewer 1
Attached is the rebuttal letter. Thank you very much for your thoughtful review of our work, and for each one of your comments.
Without any further,
The authors

Reviewer 2 Report
This paper expanded the previously established individual quality of life measurement UNCRPD with flexible change agent so that the model is more fluid to fit in different cultural/group environment, such as in Spain. The adjusted tool is design to access QoL in multiple social levels including the microsystem, the mesosystem and the macrosystem. I believe this study is a great step in transition from a theory to clinical practice in generating questions that are adapted accordingly. I only have one question/suggestion below:
In step 2 to identify specific QoL indicators, the authors suggesting using certain well-known academic databases to find out culture/group specific ones. I wonder if it is helpful adding on databases in local languages (such as in Spanish) to be even more locally specific.
Author Response
Dear Reviewer 2
Attached is the rebuttal letter. Thank you very much for your thoughtful review of our work, and for each one of your comments.
Without any further,
The authors
